# A Causal Ordering Prior for Unsupervised Representation Learning

## Abstract

Unsupervised representation learning with variational inference relies heavily on independence assumptions over latent variables. Causal representation learning (CRL), however, argues that factors of variation in a dataset are, in fact, causally related. Allowing latent variables to be correlated, as a consequence of causal relationships, is more realistic and generalisable. So far, provably identifiable methods rely on: auxiliary information, weak labels, and interventional or even counterfactual data. Inspired by causal discovery with functional causal models, we propose a fully unsupervised representation learning method that considers a data generation process with a latent additive noise model (ANM). We encourage the latent space to follow a causal ordering via loss function based on the Hessian of the latent distribution.

## 1 Introduction

The objective of extracting meaningful representations from unlabelled data is a longstanding pursuit in the field of deep learning (Bengio et al., 2013). Conventionally, methods of unsupervised representation learning have concentrated on unveiling statistically independent latent variables (Higgins et al., 2017; Chen et al., 2016; Träuble et al., 2021; Liu et al., 2022; Higgins et al., 2022), demonstrating appreciable success in synthetic benchmarks and datasets where generation parameters can be carefully manipulated (Locatello et al., 2019). However, it is essential to acknowledge the differences between controlled environments and real-world scenarios. In the latter, the factors contributing to data variation are often intertwined within causal relationships. Therefore, it is not merely advantageous but imperative to integrate causal understanding into the process of learning representations (Schölkopf et al., 2021), which can improve the models from a generalisation, and interpretability, viewpoint.

The main challenge in learning meaningful and disentangled latent representations is identifiability, i.e. ensuring the true distribution of a data generation process can be learned (up to a simple transformation, given the inherent limitation that we can never observe the hidden latent factors from observational data alone), implying the model to be injective (one-to-one mapping) onto the observed distribution. Identifiability ensures that if an estimation method perfectly fits the data distribution, the learned parameters will correspond to the true generative model. For example, discovering independent sources of variation which are observed via a nonlinear mixing function is impossible (Hyvärinen & Pajunen, 1999). This established result from the nonlinear ICA literature has been replicated for disentangled representation learning (Locatello et al., 2019).

Representation learning becomes identifiable when non-i.i.d. (independent and identically distributed) samples from a given data generation process are considered (Khemakhem et al., 2020a; Hyvärinen et al., 2023). For instance, temporal contrastive learning (Hyvärinen & Morioka, 2016) and iVAE (Khemakhem et al., 2020a) can provably ensure identifiability by utilising knowledge of auxiliary information. Indeed, Khemakhem et al. (2020a) develops a comprehensive proof that generative models become identifiable when variables in the latent space are conditionally independent, given the auxiliary information. Conditional independence given external information allows variables to be dependent (or correlated) (Khemakhem et al., 2020b), which is more realistic. Further reinforcing the notion of dependence between latent variables, the identifiability of unsupervised representations can be proven by assuming a latent space to follow a Gaussian Mixture Model (GMM) and an injective decoder (Kivva et al., 2022). Any distribution can be approximated by a mixture model

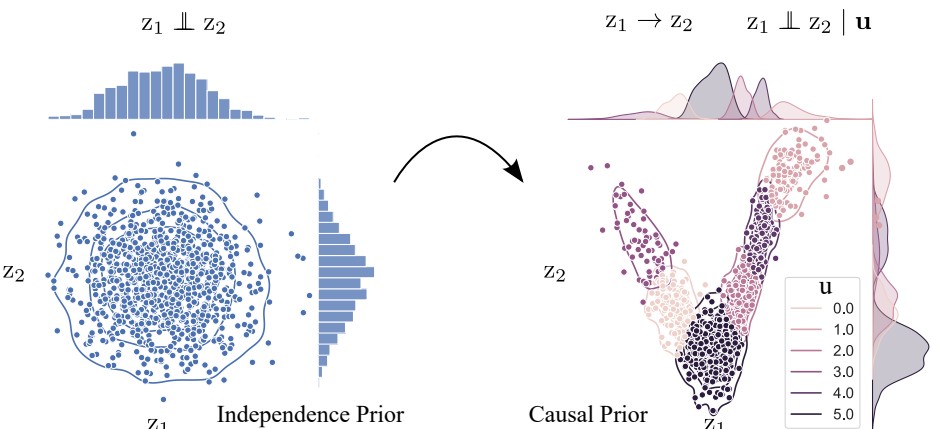

Figure 1: [Left] Independence assumption used in previous work for disentangled representations such as $\beta$-VAE and extensions. [Right] We propose to model causally related latent variables. CRL is made possible by using a mixture model in the latent space which approximates a structural causal model (SCM) with functional constraints. $z_1, z_2$ are latent variables, and $\mathbf{u}$ correspond to mixture components.

with sufficiently many components, including distributions following a causal model. The mixture component can correspond to using a "learned" auxiliary variable (Willetts & Paige, 2021), bridging the gap with (Khemakhem et al., 2020a).

Previous work (Hyvärinen & Morioka, 2016; Khemakhem et al., 2020a;b; Willetts & Paige, 2021; Hyvärinen et al., 2023) on identifiable representation learning from observational data do not consider latent causal structure. They build up, however, a theory around identifiable representation learning which allows arbitrary distribution encoding statistical dependencies in latent variables. Discovering the dependency structure in the latent space is at the core of causal representation learning (CRL) (Schölkopf et al., 2021) via the *common cause principle*[1] (Reichenbach, 1956). Learning causally related variables enable (i) robustness to distribution shifts via the independent causal mechanism (ICM) principle; (ii) better generalisation, e.g. in transfer learning settings; (iii) answering causal queries, i.e. estimation of interventional and counterfactual distributions. Previous work on CRL, however, utilises data from interventional (Ahuja et al., 2022; Varici et al., 2023) or counterfactual (pre- and post-intervention) (Locatello et al., 2020; Brehmer et al., 2022; Lippe et al., 2022) distributions for learning identifiable causal representations.

**Contributions.** In this work, we propose the COVAE (causally ordered Variational AutoEncoder) and bridge the gap between identifiable representation learning from observational data and CRL by using functional constraints (common in causal discovery (Peters et al., 2017)). We propose an unsupervised CRL method which enables drawing causal insights, from the learned latent representations. This can be done by assuming a data generation process in which the latent space adheres to an additive noise model (ANM) and applies an injective nonlinear mapping to generate observational data. In summary, the main contributions in this work include:

(i). We propose an estimation method that encourages causal ordering in the latent space, allowing us to draw causal insights from representations;

(ii). We introduce the notion stronger equivalence class ($\sim_\tau$ - *permutational block diagonal equivalence*) for model with causally ordered latent representations;

(iii). We provide theoretical results on $\sim_\tau$ −identifiability, and demonstrate the effectiveness of COVAE of multiple datasets.

---

[1]"If two observables $X$ and $Y$ are statistically dependent, then there exists a variable $Z$ that causally influences both and explains all the dependence in the sense of making them independent when conditioned on $Z$. As a special case, $Z$ can coincide with $X$ or $Y$."

## 2 RELATED WORKS

Table 1 describes data and latent space assumptions of previously existing models in comparison to the proposed method.

Table 1: Comparison of assumptions for identifiability. We describe methods by data: observational (*obs*), interventional (*int*) or counterfactual (*ctf*); and latent assumptions: independent (*ind*), conditionally independent (*cond ind*), auxiliary information (*aux*) or structural causal model (*SCM*).

| Method | Data | Latents |
|---|---|---|
| ADA-GVAE (Locatello et al., 2020) | ctf | ind |
| IVAE (Khemakhem et al., 2020a) | obs + aux | cond ind \| aux |
| VADE (Jiang et al., 2017; Willetts & Paige, 2021); MFC-VAE (Falck et al., 2021; Kivva et al., 2022) | obs | cond ind \| learned aux |
| CAUSALVAE (Yang et al., 2021), DEAR (Shen et al., 2022) | obs + aux | SCM |
| (Ahuja et al., 2022), (Varici et al., 2023) | int | SCM |
| ILCM (Brehmer et al., 2022), CITRIS (Lippe et al., 2022) | ctf | SCM |
| Ours (COVAE) | obs | SCM (ANM) |

**Disentangled Representation Learning.** Early efforts on unsupervised representation learning focused on the Variational Autoencoder framework (Kingma & Welling, 2013). $\beta$-VAE (Higgins et al., 2017) and extensions (Kim & Mnih, 2018; Eastwood & Williams, 2018; Mathieu et al., 2019) rely on independence assumptions between latent variables to learn disentangled representations (Liu et al., 2022; Higgins et al., 2022). Despite showing some success, learning independent (disentangled) representations from i.i.d. data in an unsupervised manner is provably impossible (Hyvärinen & Pajunen, 1999; Locatello et al., 2019). More recently, it was found that restricting the class of the mixing (decoder) functions to conformal maps (Buchholz et al., 2022) or volume-preserving transformations (Yang et al., 2022) results in identifiable models. Contrary to initial disentanglement works, we argue that latent variables can be causally related as illustrated in Figure 1. Here, we use injectivity constraints on the mixing function which is a weaker assumption which is possible due to our imposed latent distribution asymmetries.

**Representation Learning with Auxiliary Information.** A line of work based on nonlinear ICA leverages auxiliary information to learn identifiable models. Hyvarinen et al. (2019); Khemakhem et al. (2020a) derive a more general proof of identifiability using the concept of conditional independence given auxiliary variables. An extension of nonlinear ICA, called Independently Modulated Component Analysis (IMCA) was proposed in Khemakhem et al. (2020b), where the components are allowed to be dependent. On the contrary, Kivva et al. (2022) prove the identifiability of deep generative models can also be achieved without auxiliary information by considering a GMM prior in the latent space. In the same line, empirical results in Willetts & Paige (2021) show that the GMM prior assumption is as efficient as utilising auxiliary information in terms of learning stability (latents learned for different training seeds are correlated). We use Kivva et al. (2022) proofs as a starting point for our proofs.

**Causal Representation Learning.** It is possible to model causal relationships given access to either interventional or non-i.i.d. data. Ahuja et al. (2022) uses an injective polynomial decoder and the overall model is trained on both observational and interventional data. Varici et al. (2023) consider the case of an injective linear decoder and directly optimize the score function of the distribution (in both the latent and observation space). In Locatello et al. (2020) observations are collected before and after unknown interventions (i.e. counterfactual data), while Brehmer et al. (2022) extends this idea to causal graphs of higher complexity. Under the non-iid scenario, (Lippe et al., 2022) focuses on extracting causal factors from spatio-temporal data by performing interventions across different time steps. Works also exist that assume some level of supervision, i.e. having access to ground-truth causal factors. Shen et al. (2022) propose a GAN-based method where the prior follows a nonlinear SCM. Others (Yang et al., 2021) instead model exogenous noise directly, which is then mapped to causal latent variables via a linear SCM. Contrary to previous work, we aim at deriving causal knowledge from the latent space learning from observation data only by imposing other constraints inspired in causal discovery (Glymour et al., 2019).

## 3 DATA GENERATION PROCESS

We assume the data generation process maps the samples from latent space $\mathbf{z} \sim \mathcal{Z}$ to the samples from observational space $\mathbf{x} \sim \mathcal{O}$. $\mathbf{z}$ is a structural causal model (SCM) where each node $z_i$ depends on its parents $\mathbf{pa}(z_i)$ and some independent noise $\epsilon_i$, as illustrated in Figure 2. Formally,

$$\mathbf{x} = f_o(\mathbf{z}), \qquad\qquad p(\mathbf{z}) = \prod_i p(z_i \mid \mathbf{pa}(z_i)). \qquad (1)$$

$f_o : \mathbb{R}^d \to \mathbb{R}^o$ is a mixing function mapping latent to observation space, $d$ is the number of latent variables and $o = |\mathcal{O}| \geq d$. $\mathbf{pa}(z_i)$ are the parents of $z_i$ in $\mathcal{G}$.

**Assumption 1** (Mixing function). The mixing function $f_o$ is nonlinear piecewise affine injective function.

Under certain constraints, common neural network architectures such as multilayer perceptrons (MLPs) with LeakyRelu activation functions, follow Assumption 1. Therefore, it corresponds to a flexible and realistic class of mixing functions. We describe the constraints and propose a metric to measure injectivity of a neural network in Appendix E.

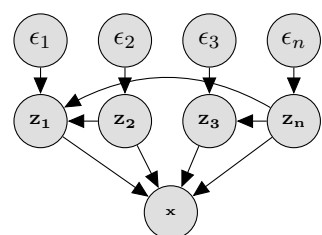

**Assumption 2** (Latent DAG). The latent distribution $p(\mathbf{z})$ is a SCM following a directed acyclic graph (DAG) $\mathcal{G}$, containing $d$ nodes, which describes the true causal structure of the latent.

**Assumption 3** (Latent Additive Noise Model, LANM). We assume that the latent SCM consists of a collection of assignments following an additive noise model (ANM) $z_i := f_i(\mathbf{pa}(z_i)) + \epsilon_i$. $\epsilon_i$ is a noise term independent of $x_i$, also called exogenous noise. $\epsilon_i$ are i.i.d. from a smooth distribution $p^\epsilon$. When using an ANM assumption over $\mathbf{z}$, the latent distribution in Equation 1 becomes

Figure 2: Data generation process with a latent SCM (endogenous and exogenous variables) causing an observation space.

$$p(\mathbf{z}) = \prod_i p(z_i \mid \mathbf{pa}(z_i)) = \prod_i p^\epsilon(z_i - f_i(\mathbf{pa}(z_i))), \qquad (2)$$

where $f_i$ is a nonlinear function and $p^\epsilon$ is any quadratic exponential noise prior (e.g. Gaussian-like) (Rolland et al., 2022; Sanchez et al., 2023).

Assuming a functional form for the causal mechanism between variables, such as ANMs (Hoyer et al., 2008; Peters et al., 2014a), is an established method for identifying causal relationships (Peters et al., 2017; Glymour et al., 2019) due to asymmetries in the joint distribution. Moreover, the ANM assumption has been shown to perform well on real benchmarks from various domains such as meteorology, biology, medicine, engineering and economy (Mooij et al., 2016), for causal discovery.

**Assumption 4** (Number of causal factors). We assume that a known number of causal factors, denoted as $d$, interact to generate the observational data $\mathbf{x}$.

**Assumption 5** ($p(\mathbf{z})$ as GMM). The latent distribution $p(\mathbf{z}) = \prod_i p^\epsilon(z_i - f_i(\mathbf{pa}(z_i))) = \sum_{j=1}^J \pi_j \mathcal{N}(\mu_j, \Sigma_j)$ can be modelled as a Gaussian Mixture Model with $J > 1$.

GMMs with a sufficient amount of components can model any densities in the limiting case (Nguyen & McLachlan, 2019). Multiple components, in turn, 'breaks the symmetry' in the latent space behaving like auxiliary information in iVAE (Willetts & Paige, 2021; Kivva et al., 2022), resulting in an identifiable model.

## 4 ENFORCING CAUSAL ORDERING IN LANM

We now derive an estimation procedure for learning the data generation process in Equation 1. We do not have access to $\mathcal{G}$ during estimation. Nevertheless, the goal is to obtain causal insights from the structure of the latent space. Therefore, we propose to encourage the latent space to be causally ordered. Causal ordering is a universal property for DAGs (Assumption 2) and therefore applicable to most causal representation learning settings. Therefore, we proceed to define what is causal order

and a loss function that will ensure that the latent space is causally ordered. Then, we describe a variational inference estimation method which models latent variables using a GMM leveraging Assumption 5.

**Definition 1** (Causal Ordering). Assume $\mathcal{G}$ to be a DAG, there is a non-unique permutation $\tau$ of $d$ nodes such that a given node always appears first in the list compared to its descendants. Formally, $\tau_i < \tau_j \iff j \in \mathbf{de}(z_i)$ where $\mathbf{de}(z_i)$ are the descendants of $z_i$ in $\mathcal{G}$ (Appendix B in Peters et al. (2017)).

### 4.1 Causal Ordering Loss

It is well known in the causal discovery literature (Glymour et al., 2019) that a complete causal graph is not identifiable from observational data without extra assumptions. If the functional form of the causal mechanism is assumed to be an ANM, causal directions become identifiable due to asymmetries.

Interestingly, previous works on causal discovery (Rolland et al., 2022; Sanchez et al., 2023) explore a property of the distribution of ANMs to find a causal ordering. The property is based on the Hessian of an ANM distribution w.r.t. its input, $\nabla^2_{\mathbf{z}_i} \log p(\mathbf{z})$. In particular, under Assumptions [2,3], $\nabla^2_{\mathbf{z}_i} \log p(\mathbf{z}) = a \iff \mathbf{z}_i$ is a leaf node, where $a$ is some constant and $\nabla^2_{\mathbf{z}_i} \log p(\mathbf{z})$ is $i^{th}$ diagonal element of the distribution's Hessian. Here, we use the same property to enforce causal ordering instead of discovering it. We encourage the Hessian of a particular node to be constant (or its variance to be zero), see Proposition 1.

**Proposition 1.** *Under Assumptions [2,3] and let $H^i_{var}(\mathbf{z}) = \mathrm{var}\left(\nabla^2_{\mathbf{z}_i} \log p(\mathbf{z})\right)$. The latent space $\mathbf{z}$ can be causally ordered by minimising the causal ordering loss defined as*

$$\mathcal{L}_{order} = -\sum_{i}^{d-1} \log \frac{H^i_{var}(\mathbf{z}_i, \ldots, \mathbf{z}_d)^{-1}}{\sum_{j=i}^{d} H^j_{var}(\mathbf{z}_i, \ldots, \mathbf{z}_d)^{-1}} \tag{3}$$

*Proof.* The proof directly extends from analysing the score of the ANM distribution

$$\nabla_{\mathbf{z}_i} \log p(\mathbf{z}) = \frac{\partial \log p^\epsilon(\mathbf{z}_i - f_i(\mathbf{pa}(\mathbf{z}_i)))}{\partial \mathbf{z}_i} - \sum_{j \in \mathbf{ch}(\mathbf{z}_i)} \frac{\partial f_j}{\partial \mathbf{z}_i} \frac{\partial \log p^\epsilon(\mathbf{z}_j - f_j(\mathbf{pa}(\mathbf{z}_j)))}{\partial \mathbf{z}_i}. \tag{4}$$

As described in Rolland et al. (2022), the minimum variance in the latent log-likelihood's hessian corresponds to a leaf node. The loss term $\mathcal{L}_{\mathrm{order}}$ is minimum if, and only if, the nodes at position $i$ are leaves. We show this by contradiction; without loss of generality, consider the random latent order $\tau$, s.t. $\tau_i \neq i$, then $H^0_{var}(\mathbf{z}) \geq \epsilon \Rightarrow \mathcal{L}_{\mathrm{order}} > 0$. Based on the above expression $\mathcal{L}_{\mathrm{order}} \to 0, \iff \tau_i = i$, where $\tau_i$ correspond to true causal order. It is important to note that as the representations are learned end-to-end, enforcing this loss would organise the latent order to follow the sorted true causal ordering. □

**Hessian Estimation.** To compute $H^i_{var}(\mathbf{z})$, we approximate the score's Jacobian (Hessian) with Stein kernel estimators (Li & Turner, 2017) as described in Rolland et al. (2022) and detailed in the Appendix B along with complexity analysis and discussion of appropriate mini-batch approximations.

---

**Algorithm 1** Compute topological loss ($\mathcal{L}_{\mathrm{order}}$)

---

1: **Initialize:** $\mathcal{L}_{\mathrm{order}} = 0, \tilde{\mathbf{K}} = \{i : \mathbf{K}\}_{i=0,\ldots d-1}, \alpha$
2: **Given:** $\mathbf{z} = f_o^{-1}(\mathbf{x})$
3: **for** $i = 0, \ldots, d-1$
4:    $\tilde{\mathbf{z}} = \mathbf{z}[i:]$
5:    $\mathbf{v} = H_{var}(\tilde{\mathbf{z}})$        ▷ Compute variance of the Hessian
6:    $\tilde{\mathbf{v}} = \mathrm{softmax}(-\log \mathbf{v})$        ▷ Smallest variance → highest $\tilde{\mathbf{v}}$
7:    $\mathcal{L}_{\mathrm{order}}+ = \mathrm{BCE}(\tilde{\mathbf{v}}, [1, 0 \ldots 0])$    ▷ First element should have smallest variance
8: **return** $\mathcal{L}_{\mathrm{order}}$

---

**Algorithmic Description.** The proposed regularization technique operates on the estimated latent representations $\mathbf{z} \in \mathbb{R}^d$. It follows an iterative process where we sequentially remove elements from $\mathbf{z}$, resulting in a modified latent representation $\tilde{\mathbf{z}} \in \mathbb{R}^{d-i}$ at each iteration $i$. During each iteration, we calculate the variance of the Hessian matrix of $\tilde{\mathbf{z}}$ with respect to the input $\mathbf{x}$. We apply a softmax activation function and compute binary cross-entropy loss to promote competition among nodes to align to a global leaf node at that iteration. This process is applied iteratively for $d-1$ iterations, effectively encouraging each element $z_j$ to be causally influenced by the nodes $z_{k>j}$.

## 4.2 Variational Inference

We are now interested in modelling a latent space with an arbitrarily complex distribution based on an ANM using the deep variational framework. That is, learning a posterior distribution that can approximate the ANM prior $p(\mathbf{z})$ given a sample from the observational distribution.

**Prior.** A multivariate diagonal Gaussian prior, as commonly used in variational autoencoders (VAE), cannot model these distributions because variables are not independent. Therefore, we consider Gaussian Mixture Model (GMM) prior under Assumption 5, following established literature (Jiang et al., 2016; Johnson et al., 2016; Falck et al., 2021), which is proven to be identifiable and have universal approximation capabilities (Kivva et al., 2022).

**ELBO.** We consider the generative model to be $p(\mathbf{x}, \mathbf{z}, \mathbf{u}) = p(\mathbf{x} \mid \mathbf{z})p(\mathbf{z} \mid \mathbf{u})p(\mathbf{u})$, following Falck et al. (2021). The posterior can be written as $q(\mathbf{u}, \mathbf{z} \mid \mathbf{x}) = q(\mathbf{u} \mid \mathbf{x})q(\mathbf{z} \mid \mathbf{x})$, where $q(\mathbf{z} \mid \mathbf{x})$ is a multivariate Gaussian with diagonal covariance and $q(\mathbf{u} \mid \mathbf{x})$ a categorical distribution over GMM components. The mixture components are inferred via prior as $q(\mathbf{u} \mid \mathbf{x}) \propto \exp(\mathbb{E}_{q(\mathbf{z}\mid\mathbf{x})} \log p(\mathbf{u} \mid \mathbf{z}))$. In this case, the posterior $q(\mathbf{u}, \mathbf{z} \mid \mathbf{x})$ is a GMM and can approximate the prior $p(\mathbf{z})$ following an ANM. A detailed derivation can be found in Appendix A.3. The ELBO for this model can be described as

$$\mathcal{L}_{\text{ELBO}} = -\mathbb{E}\left[\log p(\mathbf{x} \mid \mathbf{z})\right] + \mathbb{E}\left[\text{KL}\Big(q(\mathbf{z} \mid \mathbf{x}) \,\|\, p(\mathbf{z} \mid \mathbf{u})\Big)\right] + \text{KL}\Big(q(\mathbf{u} \mid \mathbf{x}) \,\|\, p(\mathbf{u})\Big), \quad (5)$$

where $\mathbb{E}$ is over the $q(\mathbf{u} \mid \mathbf{x})$ distribution. Based on the Proposition 1, models trained with $\mathcal{L}_{\text{total}}$ result in a causally ordered latent space $\mathbf{z}$, formally

$$\mathcal{L}_{\text{total}} = \mathcal{L}_{\text{ELBO}} + \alpha \mathcal{L}_{\text{order}} \quad (6)$$

**Discussion.** Proposition 1 shows that, given sufficient data and compute, under Assumption 3, latent representations are causally ordered. Additionally, given the organised latent representations, the causal relationships among the representations can be estimated using conditional independencies as commonly done in causal discovery (Kalisch & Bühlman, 2007; Rolland et al., 2022; Sanchez et al., 2023). The causal mechanisms between latent variables are learned implicitly.

## 5 Identifiability

A key challenge in unsupervised representation learning is identifiability. The intuition is that if two parameters result in an identical distribution of observations, then they must be equivalent in order to ensure model identifiability. Note that identifiability is the property of the data generation process, and *not* of the estimation method. Identifiability is important because it gives theoretical guarantees that an estimation method is capable of learning the true variables that generated the observed data. Formally, a data generation process resulting in a distribution $p_\theta(\mathbf{x})$ is $\sim$-identifiable up to equivalence relation $\sim$ on $\theta$, if

$$p_{\theta_1}(\mathbf{x}) = p_{\theta_2}(\mathbf{x}) \Rightarrow \theta_1 \sim \theta_2. \quad (7)$$

This exact definition of model identifiability can be too restrictive (Khemakhem et al., 2020a; Kivva et al., 2022). In reality, identifying a representation up to a simple transformation is sufficient. For example, previous work (Khemakhem et al., 2020a; Kivva et al., 2022) define a weaker form which guarantees identifiability up to affine transformation $\sim_A$ or permutation, scaling and shift $\sim_P$. In the case of an ANM data generating process, Peters et al. (2014b) demonstrates the identifiability of models with only observational data, assuming that all variables are observed. Further, Rolland et al. (2022) discuss the identifiability of ANM models under data *score* functions. However, they do not discuss the identifiability of *latent* ANM models.

In this section, we show that stronger forms of identifiability can be guaranteed when the latent ANMs are causally ordered. Firstly, we define an equivalence class considering our data generation process and estimation method. Then, we outline prior research on identifiability Kivva et al. (2022) upon which our study is built. Finally, we present our identifiability results, which goes beyond affine and permutation equivalence.

## 5.1 BACKGROUND

Recently, Kivva et al. (2022) established the identifiability of unsupervised representation learning from observational data without the need for auxiliary information. Here, we build upon their robust theoretical guarantees. However, we aim to extract causal insights from the latent space structure which was unexplored before. Thus, prior to presenting our findings, we provide an overview of their key results and establish a connection with our assumptions. We use Theorem 3.10 (a,d) in Kivva et al. (2022) which states that $f$ and $p(\mathbf{z})$ are identifiable from $p(\mathbf{x})$ up to an affine transformation ($\sim_A$ equivalence) if Assumption 1 and 5 are satisfied. Therefore, our data generation process is, at least, $\sim_A$-identifiable. We later this $\sim_A$-identifiability for proving our stronger result.

## 5.2 IDENTIFIABILITY CLASS

We now define an identifiability class which further reduces the space of transformations. As proven in Section 5.3, latent variables which are causally ordered enable stronger identifiability guarantees. The stronger guarantee derives from the fact the true causal DAG $\mathcal{G}$ can have several valid causal ordering, given the graph topology.

**Example 1.** *If $\mathcal{G}$ has $d$ nodes and no edges (independent variables), there are $d!$ possible causal orderings, since any permutation of the nodes is valid. Conversely, if the DAG is a straight line (a single path), there is only one valid causal ordering.*

**Definition 2.** (Permutational Block Diagonal Transformation, $p$) For any random variable $\mathbf{z} \in \mathcal{Z}$, a permutational block diagonal transformation is defined by $p(\mathbf{z}) = \boldsymbol{P}_\tau \cdot \mathbf{z}$ such that $\boldsymbol{P}_\tau$ is a block diagonal matrix where the blocks themselves are permutational matrices. $\boldsymbol{P}_\tau \in \mathcal{P} \subseteq \{0,1\}^{d \times d}$.

In other words, the transformation $\boldsymbol{P}_\tau$ corresponds to permutations between two valid causal ordering $\tau_i$ and $\tau_j$ of a causal graph $\mathcal{G}$. Moreover, the union of all permutation matrices between all possible causal orderings is block-diagonal, hence, block-diagonal equivalence. Computing the block size is equivalent to the maximum shift in node indices through all possible causal orderings. Finding an analytical expression for the number of causal ordering known to be $\sharp P$-complete problem Brightwell & Winkler (1991). However, we empirically show that the space of permutations between different orderings is much smaller than the space of permutations (refer Appendix D).

**Definition 3.** ($\sim_\tau$-identifiability) For $\theta = \{\mathbf{f}, \mathbf{p}\}$ a set of parameters corresponding to the mixing function and prior, the equivalence relation $\sim_\tau$ on $\theta$ is defined as:

$$(\mathbf{f}, \mathbf{p}) \sim_\tau (\tilde{\mathbf{f}}, \tilde{\mathbf{p}}) \iff \exists \quad \boldsymbol{P}_\tau \in \mathcal{P}, \boldsymbol{D} \in \mathbb{R}^{d \times d}, \mathbf{c} \in \mathbb{R}^d$$
$$s.t. \quad \mathbf{f}^{-1}(\mathbf{x}) = \boldsymbol{D} \cdot (\boldsymbol{P}_\tau \cdot \tilde{\mathbf{f}}^{-1}(\mathbf{x})) + \mathbf{c}, \forall \mathbf{x} \in \mathcal{O}, \tag{8}$$

where $\boldsymbol{P}_\tau$ is a permutational block diagonal matrix, $\boldsymbol{D}$ is a diagonal matrix for feature scaling, and $\mathbf{c}$ is a shift vector.

## 5.3 IDENTIFIABILITY OF LATENT ANMS

We prove that the latent distribution and the mixing function are identifiable under our assumptions.

**Theorem 1.** *($\sim_\tau$-identifiability of $p(\mathbf{z})$ under causal ordering) Under Assumptions [1, 2, 3, 4, 5] , $p(\mathbf{z})$ is $\sim_\tau$-identifiable from $p(\mathbf{x})$ if $\mathbf{z}$ is causally ordered.*

*Proof outline:* Based on Theorem C.2 in Kivva et al. (2022), we known that $p(\mathbf{z})$ is identifiable up to an affine transformation. With this result, we can consider $\tilde{z} = \boldsymbol{P} z + \mathbf{q} \ \forall z \sim p(\mathbf{z})$ for some invertible affine transformation $\boldsymbol{P} : \mathbb{R}^d \to \mathbb{R}^d$ and translation vector $\mathbf{q}$. Then, considering that both $\tilde{z}$ and $z$ are causally ordered, we show that $\tilde{z}, z$ can be recovered up to permutational block diagonal transformation followed by scaling and translation (indicating $\sim_\tau$ identifiability). For the complete proof, please refer to Appendix A.

*Remark* 1. In practice, we encourage the causal ordering to be a trivial sequence where the first node is a leaf (global effect), and the last node is a root (global cause).

**Theorem 2.** *(Model identifiability under causal ordering) Let $\hat{\tau}$ be the set of all possible causal ordering for the considered data distribution. Let $z \sim p(\mathbf{z})$ and $\tilde{z} \sim \tilde{p}(\mathbf{z})$, where $p(\mathbf{z})$ and $\tilde{p}(\mathbf{z})$ are latent distributions following causal ordering $\tau_p$ and $\tau_q \in \hat{\tau}$ respectively. For two invertible mixing functions $f_o, \tilde{f}_o : \mathbb{R}^d \rightarrow \mathbb{R}^{|\mathcal{O}|}$. Suppose $f_o(z), \tilde{f}_o(\tilde{z})$ are equally distributed, then there exist a linear transformation $l : \mathbb{R}^d \rightarrow \mathbb{R}^d$ and a permutational block diagonal transformation $p : \mathbb{R}^d \rightarrow \mathbb{R}^d$, such that $f_o = \tilde{f}_o \circ l^{-1} \circ p^{-1}$, indicating $f_o \sim_\tau \tilde{f}_o$.*

*Proof outline:* Given both the mixing functions $f_o, \tilde{f}_o$ are equally distributed, based on Theorem C.7 in Kivva et al. (2022), we known that there exists an invertable affine transformation $h : \mathbb{R}^d \rightarrow \mathbb{R}^d$ such that $h(z) = \tilde{z}$. Contrary to this, here we demonstrate that given causal ordering over latent factors, the affine function $h$ can be reduced to the composition of $l \circ p$. For complete proof, please refer to Appendix A.

# 6 EXPERIMENTS

In this section, we present empirical evidence showcasing the effectiveness of LANM with causal ordering constraints.

**Datasets.** We use a synthetic tabular data and image data (MorphoMNIST and Causal3DIdent datasets).

**Baselines.** We conduct a comparative evaluation of our proposed model against three baseline methods: VAE (Kingma & Welling, 2013), $\beta$-VAE (Higgins et al., 2017), and MFC-VAE (Falck et al., 2021), each employing a single facet.

**Metrics.** We compute different variants of MCC: (i) across multiple random seeds (MCC-R): measures the stability of the training process given the model; (ii) with respect to ground truth variables (MCC-GT): measures the faithfulness of the estimated latent variables to true latent variables (Khemakhem et al., 2020b); and (iii) subset MCC (MCC-SG): in the case when all parents of $\mathbf{x}$ are not observed, we measure the faithfulness by considering a subset of latent variables. As these MCC measures are permutation invariant by nature, to capture the perceived order among latent variables, we also calculate COD, which measures the divergence of the topological order in an estimated causal graph from the causal order. These metrics are formally defined

Table 2: MCC and COD results on synthetic datasets with 2, 15, and 50 nodes in the latent space along with imaging datasts MorphoMNIST-IT and MorphoMNIST-TSWI.

| METHODS($\downarrow$), METRICS($\rightarrow$) | SYN-2 | | | |
|---|---|---|---|---|
| | COD ($\downarrow$) | MCC-R($\uparrow$) | MCC-G($\uparrow$) | $R^2$($\uparrow$) |
| VAE | $0.13 \pm 0.08$ | $0.11$ | $0.26 \pm 0.03$ | $0.10 \pm 0.01$ |
| ($\beta = 0.1$)-VAE | $0.08 \pm 0.04$ | $0.14$ | $0.10 \pm 0.01$ | $0.18 \pm 0.04$ |
| ($\beta = 0.5$)-VAE | $0.11 \pm 0.08$ | $0.21$ | $0.12 \pm 0.01$ | $0.06 \pm 0.01$ |
| ($\beta = 2.0$)-VAE | $0.06 \pm 0.04$ | $0.26$ | $0.34 \pm 0.00$ | $0.11 \pm 0.00$ |
| MFC-VAE | $0.17 \pm 0.09$ | $0.14$ | $0.35 \pm 0.06$ | $0.12 \pm 0.03$ |
| COVAE | $\mathbf{0.00} \pm 0.01$ | $\mathbf{0.62}$ | $\mathbf{0.52} \pm 0.07$ | $\mathbf{0.37} \pm 0.06$ |
| | SYN-15 | | | |
| VAE | $1.68 \pm 0.22$ | $0.21$ | $0.22 \pm 0.02$ | $0.41 \pm 0.01$ |
| ($\beta = 0.1$)-VAE | $2.04 \pm 0.15$ | $0.13$ | $0.21 \pm 0.06$ | $0.38 \pm 0.04$ |
| ($\beta = 0.5$)-VAE | $1.94 \pm 0.12$ | $0.28$ | $0.18 \pm 0.04$ | $0.41 \pm 0.01$ |
| ($\beta = 2.0$)-VAE | $1.83 \pm 0.24$ | $0.24$ | $0.33 \pm 0.01$ | $0.52 \pm 0.00$ |
| MFC-VAE | $1.43 \pm 0.24$ | $0.26$ | $0.26 \pm 0.03$ | $0.48 \pm 0.08$ |
| COVAE | $\mathbf{0.03} \pm 0.01$ | $\mathbf{0.42}$ | $\mathbf{0.34} \pm 0.03$ | $\mathbf{0.56} \pm 0.05$ |
| | SYN-50 | | | |
| VAE | $5.53 \pm 0.81$ | $0.23$ | $0.28 \pm 0.24$ | $0.63 \pm 0.01$ |
| ($\beta = 0.1$)-VAE | $5.29 \pm 0.41$ | $0.11$ | $0.28 \pm 0.04$ | $0.62 \pm 0.12$ |
| ($\beta = 0.5$)-VAE | $4.15 \pm 0.35$ | $0.22$ | $0.30 \pm 0.00$ | $0.66 \pm 0.00$ |
| ($\beta = 2.0$)-VAE | $5.38 \pm 0.19$ | $0.26$ | $0.35 \pm 0.01$ | $0.66 \pm 0.00$ |
| MFC-VAE | $5.17 \pm 0.62$ | $0.31$ | $0.26 \pm 0.01$ | $0.62 \pm 0.00$ |
| COVAE | $\mathbf{0.78} \pm 0.46$ | $\mathbf{0.39}$ | $\mathbf{0.34} \pm 0.02$ | $\mathbf{0.68} \pm 0.01$ |
| METHODS($\downarrow$), METRICS($\rightarrow$) | MORPHOMNIST-IT | | | |
| | COD ($\downarrow$) | MCC-R($\uparrow$) | MCC-SG($\uparrow$) | $R^2$($\uparrow$) |
| VAE | $1.61 \pm 0.44$ | $0.29$ | $0.23 \pm 0.11$ | $0.29 \pm 0.18$ |
| MFC-VAE | $1.04 \pm 0.46$ | $0.36$ | $0.34 \pm 0.09$ | $0.42 \pm 0.16$ |
| COVAE | $\mathbf{0.0}$ | $\mathbf{0.59}$ | $\mathbf{0.47} \pm 0.08$ | $\mathbf{0.66} \pm 0.10$ |
| | MORPHOMNIST-TSWI | | | |
| VAE | $0.81 \pm 0.26$ | $0.47$ | $0.21 \pm 0.00$ | $0.24 \pm 0.04$ |
| MFC-VAE | $1.35 \pm 0.24$ | $0.52$ | $0.28 \pm 0.04$ | $0.25 \pm 0.06$ |
| COVAE | $\mathbf{0.0}$ | $\mathbf{0.61}$ | $\mathbf{0.31} \pm 0.04$ | $\mathbf{0.26} \pm 0.04$ |

logical order in an estimated causal graph from the causal order. These metrics are formally defined

in Appendix C. In addition, to quantify the injectiveness of the model we compute MIC and RRO as described in Appendix E.

## 6.1 Data Generation

**Simulation Data:**  To create the synthetic dataset, we initially generate a random latent causal Directed Acyclic Graph (DAG) with $n$ nodes and $e$ edges using the method proposed in Zhang et al. (2021). We then proceed to randomly select all the associated structural causal models $f_i$ with an *injective* mapping from $\mathbf{pa}(z_i)$ to $z_i$. Lastly, we choose an injective random transformation function $f_o$ that maps from the latent space $\mathbf{z}$ to the observational data $\mathbf{x}$. In our experimentation, we generated 2,000 data points from processes denoted as SYN-2, SYN-15, and SYN-50, where SYN-K corresponds to the aforementioned data generation process, with latent variable $\mathbf{z} \in \mathbb{R}^k$ and observational data $\mathbf{x} \in \mathbb{R}^{2k}$.

**Image Datasets:**  We also expand the applicability of our method to imaging datasets, specifically MorphoMNIST (Castro et al., 2019) variants and Causal3DIdent (Von Kügelgen et al., 2021). Concerning the MorphoMNIST dataset, we incorporate variants such as MorphoMNIST-IT, MorphoMNIST-TI, MorphoMNIST-TS, and MorphoMNIST-TSWI, where the letters I, T, S, and W correspond to latent variables $\mathbf{z}$ representing intensity, thickness, slant, and width, respectively. Detailed information about the data generation processes can be found in the Appendix. Each of the MorphMNIST variants consists of 60,000 training images and 10,000 testing images. Similarly, the Causal3DIdent dataset comprises 252,000 training samples and 25,200 test samples, all generated using a fixed causal graph with 10 nodes (additional dataset details can be found in (Von Kügelgen et al., 2021), Appendix B).

## 6.2 Results

In all our experiments, we employ a neural network model that complies with the characteristics outlined in Appendix E. Our observations, specifically with regard to the Mean Injectivity Coefficient (MIC) and Row Rank Ratio (RRO) metrics, indicate that the injectiveness of the decoder is primarily influenced by the selection of architecture and the specific dataset being analyzed. In the case of synthetic datasets, we observe the MIC of 1.0, 0.68, and 1.0 for SYN-2, SYN-15, and SYN-50 datasets, respectively, with the corresponding RRO values of 0.88, 0.93, and 0.95. Similarly, in the case of imaging datasets for both MorphoMNIST-IT and MorphoMNIST-TSWI we observe the MIC of 1.0 and RRO of 0.85. To assess the effectiveness of stability and faithfulness, we compiled in Table 2 the quantitative results.

In our analysis, we compute MCC-R using five random seeds, Table 2 illustrates the mean and standard deviation across these five runs for COD and MCC-GT. These results provide evidence that the proposed regularization, particularly in the presence of additive noise models in the latent space, effectively enforces a specific causal ordering. This is evident from the decreasing COD values approaching 0. Furthermore, based on the MCC and $R^2$ results, it can be observed that the proposed regularization also contributes to a more effective disentanglement of latent representations, improving the identifiability of the model when compared against VAE (Kingma & Welling, 2013), $\beta-$VAE (Higgins et al., 2017), and MFC-VAE (Falck et al., 2021). Additional experiments on other variants of the MorphoMNIST dataset and Causal3DIdent are detailed in the Appendix G.

## 7 Conclusion

In this work, we propose a fully unsupervised causal representation learning method for data adhering to a latent ANM by imposing a causal ordering on the latent space that corresponds to the underlying causal graph. The causal ordered latent space enables stronger identifiability results with $\sim_\tau$ equivalence. More importantly, it allows an understanding of causal ordering in the latent space. That is, a given latent variable always appears first in the latent space vector compared to its causal descendants. Possible future works would be to investigate the sample efficiency and robustness of the models trained with the proposed estimation method. Additionally, extending the proposed approach to other functional causal models and relaxing modelling assumptions and identifiability of the number of latent variables would be of particular interest.

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

# A  PROOFS

## A.1  IDENTIFIABILITY OF LATENT DISTRIBUTION UNDER CAUSAL ORDERING

Under assumptions [1, 2, 3, 5] , $p(\mathbf{z})$ is $\sim_\tau$-identifiable from $p(\mathbf{x})$ if $\mathbf{z}$ is causally ordered.

*Proof.* Let $\hat{\tau} = \{\tau_1, \dots \tau_k\}$ correspond to the set of $k$ possible causal ordering of features. Let $\boldsymbol{G}_1, \boldsymbol{G}_2 \sim \mathcal{G}$ be an adjacency graph of two samples of true DAG, following topological ordering $\tau_1, \tau_2$ modelling $y, \tilde{y}$ respectively. Given GMM's can model distribution, by breaking them into multiple piece-wise affine components, without any loss of generality we can consider:

$$z \sim p(\mathbf{z}) = \prod_i p_{\mathcal{G}}(\mathbf{z}_i \mid \mathbf{pa}(\mathbf{z}_i)) = \sum_{j=0}^{J} \pi(j)\mathcal{N}(\mu_j, \Sigma_j) \tag{9}$$

Where $\Sigma_j$ is diagonal covariance matrix, which can further be decomposed as $\Sigma_j = (\bar{\boldsymbol{A}}_j \odot \bar{\boldsymbol{A}}_j)\bar{\Sigma}$ where $\bar{\Sigma}$ is a diagonal root node covariance matrix and $\bar{\boldsymbol{A}}_j$ is diagonal scaling coefficients for that particular component. Similarly, vector $\mu_j$ can be expressed as $\mu_j = \bar{\boldsymbol{A}}_j\bar{\mu} + \bar{\mathbf{b}}_j$, where $\bar{\mu}$ is a mean vector expressed in terms of means of root node and $\bar{\mathbf{b}}_j$ is translation with respect to root nodes with respect to that component.

Let us consider a simple causal graph $x \to y$, where the mechanism $f(x)$ is non-linear (which can be modelled as piece-wise affine). For one such component where $x \in (x_0, x_1), y = ax + b$, the joint distribution, in this case, can be described using isotropic Gaussian $\mathcal{N}(\mu, \Sigma)$, where $\mu_x, \sigma_x^2$ are mean and variance of the root node. $\mu_y = a\mu_x + b$ and $\sigma_y^2 = a^2\sigma_x^2$, which can jointly described as $\mu = \boldsymbol{A}\bar{\mu}, \Sigma = (\boldsymbol{A} \odot \boldsymbol{A})\bar{\Sigma}$.

Without loss of generality consider any component $j \in \{0, \dots J\}$, resulting in covariance of $\tilde{y}$ to be:

$$\tilde{\Sigma}_j = \boldsymbol{P}\Sigma_j\boldsymbol{P}^T = \boldsymbol{P}(\bar{\boldsymbol{A}}_j \odot \bar{\boldsymbol{A}}_j)\bar{\Sigma}_j\boldsymbol{P}^T$$

Given $\tilde{\Sigma}_j, \tilde{\Sigma}_j$ are positive semi-definite (PSD), spectral decomposition of $\tilde{\Sigma}_j = \boldsymbol{V}_j\boldsymbol{V}_j^T = \boldsymbol{V}'_j\boldsymbol{V}'^T_j$, where $\boldsymbol{V}_j, \boldsymbol{V}'_j$ are PSD matrices and are unique up to orthogonal transformation $\Rightarrow \boldsymbol{V}_j = \boldsymbol{R}_j\boldsymbol{V}'_j$ for some unitary matrix $\boldsymbol{R}_j$ for each and every $j \in \{0, \dots, J\}$. Given the $\boldsymbol{G}_1$ and $\boldsymbol{G}_2$ only vary in the causal ordering, there exists a block-diagonal transformation $\boldsymbol{B}$, (transformation matrix with ones in the node indexes which belong to the same *causal hierarchy*), such that $\boldsymbol{G}_1 = \boldsymbol{B}\boldsymbol{G}_2$, this block diagonal transformation should also be reflected in the parameters of every component (given the latent variable is ordered, the mean and covariance across components also follow the same ordering), with this we can rewrite the covariances as follows:

$$(\tilde{\Sigma}_j)^{1/2} = \boldsymbol{V}_j\boldsymbol{R}_j = \boldsymbol{P}(\Sigma_j)^{1/2} = \boldsymbol{P}(\boldsymbol{B}\Sigma_j)^{1/2}$$

Without loss of generality, let's consider two components $j = 1$ and $j = 2$,

$$(\tilde{\Sigma}_1)^{1/2}(\Sigma_1)^{-1/2} = (\tilde{\Sigma}_2)^{1/2}(\Sigma_2)^{-1/2} \Rightarrow \boldsymbol{V}_1\boldsymbol{R}_1(\Sigma_1)^{-1/2} = \boldsymbol{V}_2\boldsymbol{R}_2(\Sigma_2)^{-1/2}$$

By rearranging terms, we get:

$$\boldsymbol{R}_1(\Sigma_1)^{-1/2}(\Sigma_2)^{1/2})\boldsymbol{R}_2^{-1} = \boldsymbol{V}_1^{-1}\boldsymbol{V}_2$$

Similarly, we get $\boldsymbol{V}_2^{-1}\boldsymbol{V}_1 = \boldsymbol{R}_2(\Sigma_2)^{1/2}(\Sigma_1)^{-1/2})\boldsymbol{R}_1^{-1}$ By rewriting $\Sigma_1$ in terms of $\bar{\Sigma}$ we get:

$$\boldsymbol{V}_2^{-1}\boldsymbol{V}_1 = \boldsymbol{R}_2((\bar{\boldsymbol{A}}_2 \odot \bar{\boldsymbol{A}}_2)\bar{\Sigma})^{1/2}((\bar{\boldsymbol{A}}_1 \odot \bar{\boldsymbol{A}}_1)\bar{\Sigma})^{-1/2})R_1^{-1}$$

$$\Rightarrow R_2((\bar{\boldsymbol{A}}_2 \odot \bar{\boldsymbol{A}}_2))^{1/2}\bar{\Sigma}^{1/2}\bar{\Sigma}^{-1/2}((\bar{\boldsymbol{A}}_1 \odot \bar{\boldsymbol{A}}_1)^{-1/2})\boldsymbol{R}_1^{-1}$$

$$\Rightarrow \boldsymbol{R}_2((\bar{\boldsymbol{A}}_2 \odot \bar{\boldsymbol{A}}_2))^{1/2}((\bar{\boldsymbol{A}}_1 \odot \bar{\boldsymbol{A}}_1)^{-1/2})\boldsymbol{R}_1^{-1}$$

As $\boldsymbol{R}_1, \boldsymbol{R}_2$ are unitary, $\bar{\boldsymbol{A}}_1, \bar{\boldsymbol{A}}_2$ are diagonal, PSD, and are causally ordered with respect to $G \sim \mathcal{G}$, similar to $\boldsymbol{B}$ there exists transformation matrix $\boldsymbol{B}_1, \boldsymbol{B}_2$, such that $\boldsymbol{G}_1 = \boldsymbol{B}_1\boldsymbol{G}, \boldsymbol{G}_2 = \boldsymbol{B}_2\boldsymbol{G}$. The elements in $(\bar{\boldsymbol{A}}_2 \odot \bar{\boldsymbol{A}}_2)^{1/2}(\bar{\boldsymbol{A}}_1 \odot \bar{\boldsymbol{A}}_1)^{-1/2}$ are distinct (given mixture distributions are non-degenerate and each component capture different parts of complex non-linear function), they can be uniquely

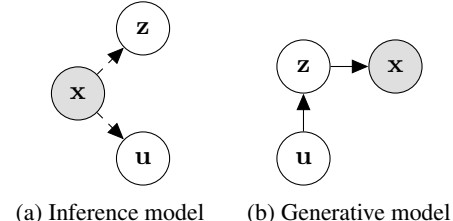

(a) Inference model    (b) Generative model

Figure 3: Variational posterior $q(\mathbf{u}, \mathbf{z} \mid \mathbf{x})$ used during inference on the left and generative model on the right. We do not give a causal interpretation for $\mathbf{c}$ in this case.

determined upto block diagonal permutation matrix $\boldsymbol{B}_1$. The spectral decomposition of $\boldsymbol{V}_2^{-1}\boldsymbol{V}_1$ results in $\boldsymbol{R}'$ such that:

$$\boldsymbol{V}_1\boldsymbol{R}'_1\boldsymbol{B}_1 = \boldsymbol{P}\Sigma_1^{1/2}, \text{ for } \boldsymbol{P}' := \boldsymbol{V}_1\boldsymbol{R}'_1, \quad \text{we have } (\boldsymbol{P}')^{-1}\boldsymbol{P} = \boldsymbol{B}_1(\Sigma_1)^{-1/2}$$

Based on this, we can conclude that, given the latent representations that follow certain causal graphs, GMMs can be identifiable up to scaling and translation (captured by the mean of components in the mixture models). $\qquad\square$

**Corollary 3.** *In the case when the causal graph is known, permutational block diagonal matrix $\boldsymbol{B}_p$ reduces to identity, giving us a scale and translation equivalence.*

If the correct causal DAG is known, the block permutation matrix $\boldsymbol{B}_p$ in theorem 1 trivially reduces to identity, resulting in scaling and translation equivalence, much stronger than affine or permutation equivalence.

## A.2   DECODER IDENTIFIABILITY UNDER CAUSAL ORDERING

Let $\hat{\tau}$ be the set of all possible causal ordering for the considered data distribution. Let $z \sim p(\mathbf{z})$ and $\tilde{z} \sim \tilde{p}(\mathbf{z})$, where $p(\mathbf{z})$ and $\tilde{p}(\mathbf{z})$ are latent distributions following causal ordering $\tau_p$ and $\tau_q \in \hat{\tau}$ respectively. For two invertible mixing functions $f_o, \tilde{f}_o : \mathbb{R}^d \to \mathbb{R}^{|\mathcal{O}|}$. Suppose $f_o(z), \tilde{f}(\tilde{z})$ are equally distributed, then there exist a linear transformation $l : \mathbb{R}^d \to \mathbb{R}^d$ and a permutational block diagonal transformation $p : \mathbb{R}^d \to \mathbb{R}^d$, such that $f_o = \tilde{f}_o \circ l^{-1} \circ p^{-1}$.

*Proof.* Given both the mixing functions $f_o, \tilde{f}_o$ are equally distributed, by Theorem C.7 Kivva et al. (2022), we know that there exists an invertable affine transformation $h : \mathbb{R}^d \to \mathbb{R}^d$, such that $h(z) := \tilde{z}$.

Based on our assumption that both distributions $p, \tilde{p}$ only vary in there partial order and the theorem 1, we can reduce the affine function as a decomposition of linear and permutation transformation, resulting in $(l \circ p)(z) = \tilde{z}$, for some invertible linear function $l : \mathbb{R}^d \to \mathbb{R}^d$ and invertible permutation function $p : \mathbb{R}^d \to \mathbb{R}^d$.

Based on the above formulation we have $f_o(z) \sim (\tilde{f}_o \circ l \circ p)(z)$, which can be rewritten as $z \sim (p^{-1} \circ l^{-1} \circ \tilde{f}_o^{-1} \circ f_o)(z)$.

If both mixing functions are equally distributed $(\tilde{f}_o^{-1} \circ f_o)(z)\forall \sim p(\mathbf{z})$ correspond to $\tilde{p}(\mathbf{z})$. This implies, based on theorem 1, $(\tilde{f}_o^{-1} \circ f_o) \equiv (l' \circ p')$ for some random linear and permutation functions $l'$ and $p'$ respectively.

This results in $(p^{-1} \circ l^{-1} \circ \tilde{f}_o^{-1} \circ f_o) = (l' \circ p')$ on domain $f_o^{-1}(\mathcal{O})$.

We get $\tilde{f}_o(\tilde{z}) = (f_o \circ l' \circ p')(z) \quad \forall \quad z \in \tilde{f}_o^{-1}(\mathcal{O})$ $\qquad\square$

## A.3   ELBO DERIVATION

We now derive the ELBO used in this work which follows Falck et al. (2021).

For this, we start with the data distribution as $p(\mathbf{x})$, and the aim is to maximize the log-likelihood of this distribution:

$$\log p(\mathbf{x})$$

$$= \log \int_{\mathbf{u}} \int_{\mathbf{z}} p(x, \mathbf{u}, \mathbf{z}) d\mathbf{z} d\mathbf{u}$$

Let's consider variational distributions $q(\mathbf{u}, \mathbf{z} \mid \mathbf{x})$.

$$= \log \int_{\mathbf{u}} \int_{\mathbf{z}} p(\mathbf{x}, \mathbf{u}, \mathbf{z}) \frac{q(\mathbf{u}, \mathbf{z} \mid \mathbf{x})}{q(\mathbf{u}, \mathbf{z} \mid \mathbf{x})} d\mathbf{z} d\mathbf{u}$$

$$\geq \mathbb{E}_{q(\mathbf{u}, \mathbf{z} \mid \mathbf{x})} \log \frac{p(\mathbf{x}, \mathbf{u}, \mathbf{z})}{q(\mathbf{u}, \mathbf{z} \mid \mathbf{x})}$$

Based on modelling assumption described in Figure 3, $q(\mathbf{u}, \mathbf{z} \mid \mathbf{x})$ decomposes as $q(\mathbf{u} \mid \mathbf{x})q(\mathbf{z} \mid \mathbf{x})$

$$= \mathbb{E}_{q(\mathbf{u}, \mathbf{z} \mid \mathbf{x})} \left[ \log p(\mathbf{x} \mid \mathbf{z}) + \log \frac{p(\mathbf{u})}{q(\mathbf{u} \mid \mathbf{x})} + \log \frac{p(\mathbf{z} \mid \mathbf{u})}{q(\mathbf{z} \mid \mathbf{x})} \right]$$

$$= \mathbb{E}_{q(\mathbf{z} \mid \mathbf{x})} \log p(\mathbf{x} \mid \mathbf{z}) + \mathbb{E}_{q(\mathbf{z} \mid \mathbf{x})} \mathbb{E}_{q(\mathbf{u} \mid \mathbf{x})} \log \frac{p(\mathbf{u} \mid \mathbf{z})}{q(\mathbf{u} \mid \mathbf{x})} + \mathbb{E}_{q(\mathbf{z} \mid \mathbf{x})} \log \frac{p(\mathbf{z})}{q(\mathbf{z} \mid \mathbf{x})}$$

$$= \mathbb{E}_{q(\mathbf{z} \mid \mathbf{x})} \log p(\mathbf{x} \mid \mathbf{z}) - \mathbb{E}_{q(\mathbf{z} \mid \mathbf{x})} \mathrm{KL}\Big(q(\mathbf{u} \mid \mathbf{x}) \| p(\mathbf{u})\Big) - \mathrm{KL}\Big(q(\mathbf{z} \mid \mathbf{x}) \| p(\mathbf{z} \mid \mathbf{u})\Big)$$

$$\Rightarrow \mathcal{L}_{\mathrm{ELBO}} = -\mathbb{E}_{q(\mathbf{z} \mid \mathbf{x})} \log p(\mathbf{x} \mid \mathbf{z}) + \mathrm{KL}\Big(q(\mathbf{u} \mid \mathbf{x}) \| p(\mathbf{u})\Big) + \mathbb{E}_{q(\mathbf{u} \mid \mathbf{x})} \mathrm{KL}\Big(q(\mathbf{z} \mid \mathbf{x}) \| p(\mathbf{z} \mid \mathbf{u})\Big)$$

## B  HESSIAN ESTIMATION

To compute $H_{var}^i(\mathbf{z})$, we approximate the score's Jacobian (Hessian) with Stein kernel estimators (Li & Turner, 2017) as described in Rolland et al. (2022) and detailed in the Appendix:

$$\mathbf{J}^{Stein} = -\mathrm{diag}(\mathbf{G}^{Stein}(\mathbf{G}^{Stein})^T) + (\mathbf{K} + \eta\mathbf{I})^{-1}\langle\nabla^2_{diag}, \mathbf{K}\rangle \tag{10}$$

Where $\mathbf{G}^{Stein} = -(\mathbf{K} + \eta\mathbf{I})^{-1}\langle\nabla, \mathbf{K}\rangle$ is the Stein gradient estimator Li & Turner (2017), $\mathbf{K}$ is the median kernel, $\mathbf{I}$ is the Identity matrix, and $\langle a, b\rangle$ correspond to applying operation $a$ on $b$ element-wise. The final algorithm for computing $\mathcal{L}_{\mathrm{order}}$ is described in Alg. 1.

**Complexity analysis.** As outlined in Algorithm 1, our proposed framework includes two main complexity-inducing steps (i) Jacobian estimation (line 5 of algorithm 1): for this we use kernel-based estimation method detailed in Equation 10, which requires inverting $b \times b$ matrix ($b$ is the batch size used) resulting in an additional complexity of $O(b^3)$, and (ii) the loop over all latent variables (line 3 in Alg. 1): this further increases the factor of complexity resulting in $O(db^3)$. The complexity can be reduced by the heuristic of causally ordering top $m$ variables, where $m << d$, resulting in the final complexity of $O(mb^3)$.

**Kernel estimation to mini-batch approximation.** The stein estimator in Equation 10 is a kernel-based approach, which means it requires an entire data distribution to compute jacobian, here we approximate it using mini-batch optimization while preserving the kernel characteristics. For this, we consider the moving average over kernel statistics across batches, which eventually converges to entire dataset statistics.

---

**Algorithm 2** Compute variance of the Hessian ($H_{var}(\mathbf{z})$)

---

1: **Given:** $\mathbf{z} = f^{-1}(\mathbf{x})$
2:   $\tilde{\mathbf{K}}[i] = (1 - \alpha)\tilde{\mathbf{K}}[i] + \alpha \mathbf{K}(z)$
3:   **Compute:** $\mathbf{G}^{Stein}(\tilde{z}, \tilde{\mathbf{K}}[i])$                    ▷ Compute gradient
4:   $\mathbf{v} = \text{var}\left(\mathbf{J}^{Stein}(\mathbf{G}^{Stein}, z, \tilde{\mathbf{K}}[i])\right)$       ▷ Compute variance of a Jacobian of a score

---

## C    METRICS

We compute different variants of MCC: (i) across multiple random seeds (MCC-R): measures the stability of the training process given the model; (ii) with respect to ground truth variables (MCC-GT): measures the faithfulness of the estimated latent variables to true latent variables Khemakhem et al. (2020b); and (iii) subset MCC (MCC-SG): in the case when all parents of $\mathbf{X}$ are not observed, we measure the faithfulness by considering a subset of latent variables. All three variants are formally described in definition 4. As these MCC measures are permutation invariant by nature, to capture the perceived order among latent variables, we also calculate COD, which measures the divergence of the topological order in an estimated causal graph from the causal order, formally defined in equation 13. In addition, to quantify the injectiveness of the model we compute MIC and RRO defined in 6.

**Definition 4.** (Mean Correlation Coefficient) We compute the mean correlation coefficient with respect to ground truth (MCC-G) as described in Khemakhem et al. (2020b). MCC-SG and MCC-R are based on MCC-G and are described as:

$$\text{MCC-SG}(\hat{\mathbf{z}}, \mathbf{z}) = \max\left\{\text{MCC-G}(\hat{\mathbf{z}}[S_j], \mathbf{z}), \quad \forall j = \{1, \ldots, |\mathcal{S}|\}, \quad S = \binom{|\hat{\mathbf{z}}|}{|\mathbf{z}|}\right\} \quad (11)$$

$$\text{MCC-R}(\{\hat{z}_0, \ldots, \hat{z}_K\}) = \frac{1}{K-1}\sum_k \text{MCC-G}(\hat{z}_k, \hat{z}_0), \quad (12)$$

where $\hat{\mathbf{z}}_k = \mathbf{f}_k^{-1}(\mathbf{X})$, $S$ is the set of all the partition indices of $\hat{z}$ with the size of $|\mathbf{z}|$, $\mathbf{z}$ corresponds to the ground truth latent features and $K$ total number of experimental runs.

**Definition 5.** (Causal Order Divergence, COD) Similar to divergence metric in Rolland et al. (2022); Sanchez et al. (2023), we define COD as:

$$\text{COD}(\tau, A) = \sum_{i=0}^{d}\sum_{j>i}^{d} A_{ij} \quad (13)$$

where $\tau = \{0, \ldots, d\}$ is the expected order and $A$ is an estimated adjacency graph predicted using the resulting latent space after training.

## D    EMPIRICAL ANALYSIS OF EQUIVALENCE CLASS

Here, we empirically analyse the benefits of stronger block diagonal transformation in reducing search space. For this, we randomly generated a DAG as illustrated in Figure 4(a). Our results show that, on average, at most (depending on the number of nodes), 1% of all permutations are possible causal orderings. Figure 4(b) demonstrates all possible causal ordering for the considered DAG in Figure 4(a), it can be observed that all possible permutation for this particular graph is 8!, while selecting between a set of causal ordered is just 14. The graph in Figure 5 demonstrates the search space ratio as the number of nodes and edges increases in the graph.

## E    NEURAL NETWORK CONSTRAINTS FOR INJECTIVE DECODERS

It is common to assume an injective decoder (mixing function) for proving the identifiability of a data generation process Kivva et al. (2022). When implementing a deep generative model in

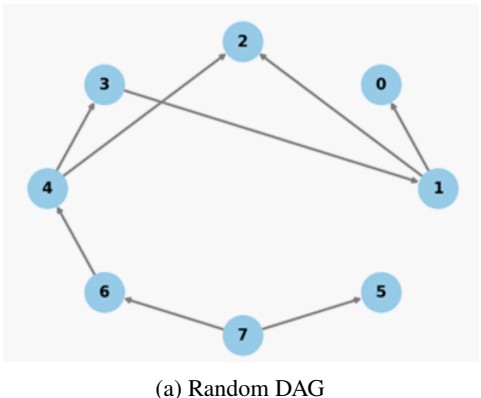

| All possible causal ordering | | | | | | | |
|---|---|---|---|---|---|---|---|
| 7 | 6 | 4 | 3 | 1 | 2 | 0 | 5 |
| 7 | 6 | 4 | 3 | 1 | 2 | 5 | 0 |
| 7 | 6 | 4 | 3 | 1 | 0 | 5 | 2 |
| 7 | 6 | 4 | 3 | 1 | 0 | 2 | 5 |
| 7 | 6 | 4 | 3 | 1 | 5 | 2 | 0 |
| 7 | 6 | 4 | 3 | 1 | 5 | 0 | 2 |
| 7 | 6 | 4 | 3 | 5 | 1 | 2 | 0 |
| 7 | 6 | 4 | 3 | 5 | 1 | 0 | 2 |
| 7 | 6 | 4 | 5 | 3 | 1 | 2 | 0 |
| 7 | 6 | 4 | 5 | 3 | 1 | 0 | 2 |
| 7 | 6 | 5 | 4 | 3 | 1 | 2 | 0 |
| 7 | 6 | 5 | 4 | 3 | 1 | 0 | 2 |
| 7 | 5 | 6 | 4 | 3 | 1 | 2 | 0 |
| 7 | 5 | 6 | 4 | 3 | 1 | 0 | 2 |

(a) Random DAG         (b) Causal Ordering

Figure 4: Figure illustrates a random DAG and its all corresponding causal orders

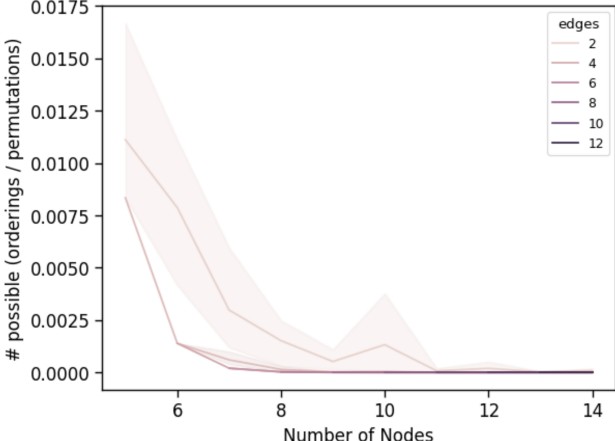

Figure 5: Figure illustrates the ratio between the number of causal orders and total number of permutations

practice, some constraints in the decoder are necessary to ensure that neural networks are modelling injective functions. We follow similar modelling assumptions of ICE-BeeM Khemakhem et al. (2020b): (i) Monotonicity: The latent dimension of the decoder is monotonically increasing, *i.e.,* $d_{l+1} \geq d_l \quad \forall l \in \{0, \ldots, L-1\}$, where $d_l$ corresponds to the feature dimension at layer $l$ and $L$ is the total number of layers in the decoder. (ii) Activation: The activation function after every layer corresponds to LeakyReLU ($\max(0,x) + \alpha \min(0,x), \alpha \in (0,1)$). (iii) Full rank: All weight matrices $\mathbf{f}_l$ are full row ranked, as the number of rows is greater than or equal to the number of columns. (iv) Invertible sub-matrix: All weight sub-matrices $\mathbf{f}'_l$ of size $d_l \times d_l$ are invertible.

Based on the network constraints described above, we propose MIC, a measure of *injectivity* of the model of the resulting model (after training).

**Definition 6.** (Mean Injectivity Coefficient, MIC) MIC is formally described as

$$\text{MIC}(\mathbf{f}) = \min \left\{ \frac{1}{|\mathcal{C}|} \sum_j \frac{Rank(\mathbf{f}_i(\mathcal{C}_j)^T)}{ri} \quad \forall j \in \{0, \ldots, |\mathcal{C}|\} \right\} \tag{14}$$

where, $ci, ri$ correspond to number of columns and rows of $\mathbf{f}_i$, with abuse of notation, we use $\mathcal{C} = \binom{ci}{ri}$ as a set of all partitions of column indices with size $ri$, and $|S|$ is the cardinality of set $S$.

*Remark* 2. We measure the average row rank ratio RRO $= \left( \frac{1}{L} \sum_l \frac{Rank(f_l)}{d_l} \right)$ and MIC (ref. equation 14) to measure the injectivity of the decoder.

# F    EXPERIMENTAL SETUP

## F.1    DATA GENERATING PROCESS - MORPHOMNIST DATASET

Here, we synthetic data based on MNIST digits Castro et al. (2019). We define multiple data-generating process with four different variables thickness, width, slant, and intensity, and evaluate our proposed method in terms of MCC's and COD. Here, thickness corresponds to the stroke thickness of a digit, width corresponds to the total width of a written digit, slant corresponds to the shear factor along a horizontal direction, and intensity corresponds to the average intensity of pixels in a digit. Functions $SetIntensity(x; i)$, $SetSlant(x; s)$, $SetWidth(x; w)$, and $SetThickness(x; t)$ refer to the operations applied to the original MNIST digit to generate new image $x$ with desired properties by controlling image morphology. We use the data-generating process similar to the ones described in Kori et al. (2022), we formally describe them below.

**Morpho-MNIST-TI**: In this setting we consider two causal variables thickness and intensity, where thickness causes intensity. Mathematically the functional relationship between variables are defined as described in equation 15.

$$
\begin{aligned}
t &:= f_t \triangleq 0.5 + \epsilon_t \quad \epsilon_t \sim \Gamma(10, 5) \\
i &:= f_i \triangleq 64 + 191 * \sigma(2 * w + 5) + \epsilon_i \quad \epsilon_i \sim \mathcal{N}(0, 1) \\
x &:= f_x = SetIntensity(SetThickness(X; t); i)
\end{aligned}
\tag{15}
$$

**Morpho-MNIST-IT**: In this experiment we inverted a directionality from previous setting resulting in intensity to cause thickness, which is mathematically described in equation 16

$$
\begin{aligned}
i &:= f_i \triangleq \epsilon_i \quad \epsilon_i \sim \mathbb{U}(60, 255) \\
t &:= f_t \triangleq 3 + \sigma(i/255) + \epsilon_s \quad \epsilon_s \sim \mathcal{N}(0, 0.5) \\
x &:= f_x = SetThickness(SetIntensity(X; i); t)
\end{aligned}
\tag{16}
$$

**Morpho-MNIST-TS**: In this setup we use thickness and slant as causal attributes, where thickness causes digit slantness, which is formally described in equation 17

$$
\begin{aligned}
t &:= f_t \triangleq \epsilon_t \quad \epsilon_t \sim \Gamma(0, 5) \\
s &:= f_s \triangleq 10 + 5 * \sigma(2 * t - 5) + \epsilon_s \quad \epsilon_s \sim \mathcal{N}(0, 0.5) \\
x &:= f_x = SetSlant(SetThickness(X; t); s)
\end{aligned}
\tag{17}
$$

**Morpho-MNIST-TSWI**: In this setup we increased a complexity by using intensity, thickness, slant, and digit width as a causal attributes, where thickness causes slant, thickness and slant causes width, and width causes intensity. This data-generating process is formally described in equation 18

$$
\begin{aligned}
t &:= f_t \triangleq \epsilon_t \quad \epsilon_t \sim \Gamma(0,5) \\
s &:= f_s \triangleq 10 + 20 * t + \epsilon_s \quad \epsilon_s \sim \mathcal{N}(0,5) \\
w &:= f_w \triangleq 10 + 15 * \sigma(0.5 * t) - 0.25 * s + \epsilon_w \quad \epsilon_w \sim \mathcal{N}(0,1) \\
i &:= f_i \triangleq 64 + 191 * \sigma(w/25) + \epsilon_i \quad \epsilon_i \sim \mathbb{N}(0,1) \\
x &:= f_x = SetIntensity(SetWidth(SetSlant(SetThickness(X;t);s);w);i)
\end{aligned} \tag{18}
$$

### F.2 CODE AND IMPLEMENTATION

We use the latent GMM loss from MFC-VAE Falck et al. (2021) inspired in the implementation from `https://github.com/FabianFalck/mfcvae`. We also append the code for the model and loss functions used in the paper to the supplemental material.

### F.3 HYPERPARAMETERS

In Table 3 we detail all the hyper-parameters used in our experiments. We use a fixed decoder standard deviation in the case of CAUSAL3DIDENT and MORPHOMNIST, while in the case of SYN-K dataset it remains learnable (described as $\sigma$ in the table). It is also worth mentioning that for the VAE method on CAUSAL3DIDENT, we trained a deeper model and also set the KL weight term $\beta$ equal to 0 to ensure fair comparison with the other two methods and avoid posterior collapse, respectively.

## G RESULTS

Table 4 depicts final results on MORPHOMNIST-TI, MORPHOMNIST-TS, and CAUSAL3DIDENT dataset, respectively. For each method, we re-run all experiments and collect metrics across 5 different random seeds for MORPHOMNIST-TI and MORPHOMNIST-TS, and 3 random seeds for CAUSAL3DIDENT. For the latter dataset, we observed that all three metrics exhibit high variance across runs; however, it is clear that both MFC-VAE and COVAE are comparable methods.

Table 3: Experimental details w.r.t models and datasets

| DATASETS($\downarrow$),
METHODS($\rightarrow$) | | VAE | MFC-VAE | COVAE |
|---|---|---|---|---|
| | No. Layers | | 3 if k < 3 else 6 | |
| | Training Steps | | 15600 | |
| | No. Samples | | 2000 | |
| | Batch Size | | 256 | |
| SYN-K | Optimizer | | Adam | |
| | Learning Rate | | 5e-4 | |
| | $\alpha$ | - | 0.0 | 1.0 |
| | $\beta$ | 1.0 | 1.0 | 1.0 |
| | Decoder $\sigma$ | | $\sigma$ | |
| | No. Layers | | 6 | |
| | Training Steps | | 6000 | |
| | No. Samples | | 60000 | |
| | Batch Size | | 256 | |
| MORPHOMNIST | Optimizer | | Adam | |
| | Learning Rate | | 1e-4 | |
| | $\alpha$ | - | 0.0 | 1.0 |
| | $\beta$ | 1.0 | 1.0 | 1.0 |
| | Decoder $\sigma$ | 0.5 | 0.5 | 0.5 |
| | Input resolution | | $64 \times 64$ | |
| | No. Layers | 4 | 3 | 3 |
| | Training Steps | | 19687 | |
| | No. Samples | | 252000 | |
| | Batch Size | | 128 | |
| CAUSAL3DIDENT | Optimizer | | Adam | |
| | Learning Rate | | 5e-4 | |
| | Hidden dim | | 256 | |
| | Latent dim | 256 | 16 | 16 |
| | $\alpha$ | - | 1.0 | 1.0 |
| | $\beta$ | 0.0 | 0.01 | 0.01 |
| | Decoder $\sigma$ | 0.1 | 0.1 | 0.1 |

Table 4: MCC and COD results on MorphoMNIST and Causal3DIdent datasets

| METHODS($\downarrow$),
METRICS($\rightarrow$) | MORPHOMNIST-TI | | |
|---|---|---|---|
| | COD ($\downarrow$) | MCC-R($\uparrow$) | MCC-SG($\uparrow$) |
| VAE | $1.31 \pm 0.28$ | 0.31 | $0.24 \pm 0.06$ |
| MFC-VAE | $1.33 \pm 0.38$ | 0.38 | $\mathbf{0.39} \pm 0.07$ |
| COVAE | $\mathbf{0.0}$ | $\mathbf{0.58}$ | $0.38 \pm 0.06$ |
| | MORPHOMNIST-TS | | |
| VAE | $1.47 \pm 0.65$ | 0.48 | $0.38 \pm 0.05$ |
| MFC-VAE | $1.75 \pm 0.60$ | 0.51 | $0.36 \pm 0.06$ |
| COVAE | $\mathbf{0.0}$ | $\mathbf{0.56}$ | $\mathbf{0.41} \pm 0.05$ |
| | CAUSAL3DIDENT | | |
| VAE | $22.39 \pm 1.49$ | 0.15 | $0.15 \pm 0.0$ |
| MFC-VAE | $\mathbf{3.56} \pm 0.87$ | $\mathbf{0.28}$ | $\mathbf{0.27} \pm 0.01$ |
| COVAE | $3.94 \pm 0.86$ | 0.26 | $0.25 \pm 0.02$ |

