# OpenReview forum: "A Causal Ordering Prior for Unsupervised Representation Learning"
_ICLR.cc/2024/Conference — Submitted to ICLR 2024_

### Official Review · Reviewer_5aVx · 2023-10-30

**Soundness:** 1 poor
**Presentation:** 2 fair
**Contribution:** 2 fair
**Rating:** 3
**Confidence:** 5

**Summary:**

In recent years, the theory of identification for causal representation learning has come into prominence. This work proposes extensions to this theory and builds methods from the theory and empirically validates the proposal. The work makes following crucial assumptions to arrive at the identification guarantees: i) mixing map is piecewise affine, ii) the latent causal model is additive noise model, iii) the latent causal model is also expressed as a mixture of Gaussians. Under these assumptions, authors leverage two key recent results – a. Kivva et al., which establishes affine identification for gaussian mixture latents and piecewise affine mixing maps, and  b. Rolland et al., which provides a score based approach for causal discovery, to establish representation identification guarantees.

**Strengths:**

Along the axis of originality and significance, I believe the paper has some strengths. The problem that the authors have picked is a very important one. We need more work and ideas on how to move beyond standard independence assumptions on the latents. The authors considered the pure observational data setting, which is quite a challenging one.  The idea to combine the contributions from Kivva et al., and Rolland et al., to achieve identification with observational data only is interesting and practical.

**Weaknesses:**

I have some major concerns with the paper and these concerns would fall under the quality axis and clarity axis. I elaborate on these below.

1. **Issue with Proposition 1:**
The proof of Proposition 1 is not clear and seems incorrect. The loss function is summed from i=1 to d? It is not clearly written it says i to d. The term inside the summation can be undefined for non leaf nodes. The numerator in the logarithm would be non-zero but the denominator would be infinite if the denominator sums over terms that include a leaf node.
The proposition relies on results Rolland et al. However, it seems to use it incorrectly. Rolland’s result assumes that the joint variables are Markov w.r.t a certain DAG. However, for a general set of latents z that are learned such a condition need not hold to begin with.

2.  **Issue with Assumption 5:**  In Assumption 5, authors require the ANM latent model to be expressed as a mixture model. Below the assumption the authors justify it to be a relatively benign assumption as mixture models have arbitrary expressive power. However, there is a crucial point that is missed here. To express any distribution, the number of components need to go to infinity. However, the result from Kivva et al.  is only for finite mixtures.

Now that means that the result requires that latent ANM model to be expressed as a finite mixture. The authors should provide examples of the data generation process that follows this assumption. If I were to follow the proof in the Appendix and reverse engineer a DGP that is compatible with it, there are some contradictions that we run into. From equation (9), the adjacency matrix dictates the covariance matrix. From the LHS in equation (9), it follows that conditioned on the parents the variable is independent of the ancestors. From the RHS in equation (9), it is not clear if the same independence condition holds. Rest of the proof relies on equation (9) and I am unable to trust the proof due to lack of clarity. I request the authors to construct a clear data generation process.

Consider the set of latents Z1,Z2,Z3,Z4 and H. The DAG governing their relationships is Z1→ Z2→Z3→Z4  and H→Z1, H→Z2, H→Z3.  We can write an SCM for this situation as follows

Z1 <-- H + N1, Z2 <--Z1+ H + N2, Z3 <-- Z2 + H+ N3, Z4 <-- Z3 + H + N3.

If H is a discrete random variable and all other variables are Gaussians, then Z1,Z2,Z3,Z4 conditional on H=h, follows a Gaussian distribution. This is a type of SCM consistent with RHS of equation (9).  Note that in the above DAG, Z3 conditioned on Z2 is not independent of Z1 as there is a path from Z3 to Z1 through H. The LHS of equation (9) requires that each Z_i is independent of its ancestors conditioned on its parents in Z. Thus there is a contradiction.


Minor remark: In related works, you say Yang et al. show that it is possible to achieve identifiable models using volume preserving transforms. There are errors in that result -- see https://openreview.net/pdf?id=REtKapdkyI.

**Questions:**

In the weakness section, I have highlighted the key issues. The authors can respond to those.

---

### Official Review · Reviewer_4VV5 · 2023-10-31

**Soundness:** 2 fair
**Presentation:** 2 fair
**Contribution:** 1 poor
**Rating:** 3
**Confidence:** 4

**Summary:**

This paper is on learning causal representations of data in an unsupervised manner. A recent line of work in deep learning has proposed to learn data representations, either supervised or unsupervised, which are additionally causal. In this work, the representations are assumed to have a structural causal model on them. Furthermore, the authors make the same assumptions made in Kivva et al., that the latent priors form a Gaussian mixture model and that the mixing function is assumed to be non-linear piecewise affine. This model has universal approximation properties and identifiability results for such models have been shown before in (Kivva et al.). The authors extend this theory using additional assumptions (however, see weaknesses below), to a slightly better class of transformations.

To learn this, the authors propose the coVAE model to learn these representations. Using inspiration from prior work (Rolland et al.), the authors use a Hessian property to enforce acyclicity of the latent additive noise model. This lets them recover the causal ordering among the latents, and similar ideas as (Rolland et al.) are used. Experiments on synthetic datasets and simple image datasets (Causal3DIdent and variants of MorphoMNIST) validate their identifiability results (which are measured via MCC metrics and a new COD metric). The work is written well however it does not have many significant contributions either theoretically or experimentally.

**Strengths:**

- The research direction is important because causal representation learning is a promising extension of neural methods that aims to leverage additional causal information for robustness and generalization.

- The proposed method coVAE shows improved MCC and R^2 metrics over prior experimental baselines, i.e. improved recovery of meaningful latents.

**Weaknesses:**

- The abstract claims that all provable identifiable methods rely on additonal information, however the main work (Kivva et al.) that the authors cite, show that we do not need additional information, therefore, this claim should be revised.

- It's hard to understand how Assumptions 2, 3, 5 together work. They seem unrelated assumptions with different motivations -- Assumption 2, 3 on latent SCMs is more directly related to the paper since it's inherently causal; however, Assumption 5 on GMMs is needed for identifiability due to some prior works. Could the authors precisely describe what kind of GMMs satisfy this sort of SCM structure?

- The theory is incremental compared to (Kivva et al.), The main contribution of this work is that linear identifiability is strengthened to identifiability up to permutation block diagonal transformations. Could the authors comment on why this improvement is desirable in the context of latent variable models?

- Since the main thrust of this work is experimental, I'd liked to see more a more comprehensive evaluation of coVAE. For instance, additional larger datasets and also potentially ablation studies (the dependence on alpha) would sigificantly strengthen the work.

**Questions:**

Please address the questions above.

Typo: In the 2nd line of proof in A.2, "invertable" -> "invertible"

---

### Official Review · Reviewer_fvDC · 2023-10-31

**Soundness:** 2 fair
**Presentation:** 3 good
**Contribution:** 3 good
**Rating:** 5
**Confidence:** 3

**Summary:**

The paper suggests a new method to learn latent causal structures based on a more complex mixture of Gaussian prior in the latent space and a causal order loss based on the Jacobian of the score function. They study their approach on different datasets and they provide some identifiability results.

**Strengths:**

- The paper studies an important problem, i.e., how to learn latent variables with causal relations.
- It nicely combines ideas from different fields: Identifiability of latent variable model with piecewise linear mixing, Gaussian mixture models, score based causal discovery.
- The main paper is generally easy to follow
- Experimental results look generally promising.

**Weaknesses:**

- There do not seem to be any completely novel ideas in the paper (score based causal discovery, gmm latent prior, identifiability without auxiliary information). (this is a minor point)
- For the identifiability part, there are some questions regarding the combination of assumptions (see questions).
- The proofs in Appendix A could be much clearer (see questions), and, more generally, all math parts should be checked carefully (being slightly imprecise can make it very difficult to follow the details).
- There are some questions regarding the metrics used in the experiments
- References to 2023 works on CRL seem to be mostly missing. (just a todo, not relevant for the score)

**Questions:**

- Note that while GMM can approximate any distribution, the identifiability result in Kivva et al. (2022) does not hold for arbitrary distributions (i.e., the closure of GMMs). So in practice, it becomes harder to identify more complex mixtures. (I did not find any error in this direction in the paper but in the introduction it is emphasized that GMMs can approximate any distribution which might be not the right way to think about this result).

- In relation to the previous point, I think that Assumption 5 needs stronger justification. Linear Gaussian SCMs clearly fall into this class, but what about further examples? Is there a closer connection beyond the generic approximability statement?

- When reading Definition 2 it sounds like the set $\mathcal{P}$ just agrees with all permutation matrices, or is there some implicit assumption about the block structure hidden in $\tau$? (all permutation matrices are block diagonal permutation matrices with one block) This also has consequences for Definition 3.

- From a theoretical standpoint, the causal order viewpoint seems less elegant than, e.g., stating identifiability of variables and graph up to permutations (relabeling).

- Check statement of Theorem 2, if $l$ is really linear you just reduce from affine to linear, do you mean diagonal?

- I could not follow the proof in Appendix A. There seem to be substantial inaccuracies, e.g., $G_1\Sigma_j$ is generally not symmetric and therefore no covariance matrix. Also the general reasoning is not clear and a proof should not consist of 3 lines followed by a one page 'Example' paragraph. Given my understanding what could be done here is to combine existing results to conclude identifiability up to linear maps followed by an identifiability result for linear mixing function (e.g., Varici et al.). Then the only problematic point would be the SCM distribution = GMM. But this does not seem to be the case, here. It looks like some kind of GMM decomposition for an SCM is proved but this must be clarified and proved beyond a two node example. It is not clear how the affine ambiguity is removed in the proof.

- It should be clarified whether the used causal order loss is consistent when not only trying to find leaf nodes given $(Z_1,\ldots,Z_d)$
but given only linear mixtures $AZ$.

- I am a bit puzzled by the COD metric used in the experiments. My understanding is the following: You learn a latent structure, d then you apply Causal discovery on the latent space and finally evaluate the indicated COD metric? However, this seems to have substantial shortcomings as it can be tricked easily, e.g., learning a random iid. representation (like the Darmois construction) results in a COD of 0. In particular, training a VAE such that the learned embeddings match the isotropic Gaussian prior in the latent space should result in an empty causal graph (independent variables) and therefore perfect COD, even though no information about the causal structure were obtained. Under the assumption that this is correct so far, the experiments provide no evidence that the true causal structure is found but potentially the causal order loss just results in representations where no arrows in the wrong direction are found, however, the arrows do not recover the ground truth structure. More evidence is needed to show that the approach helps to learn causal structures.

- generally COD are tiny, so the estimated graphs seem very sparse (compared to the number of errors made in the simpler causal discovery problem). What method was used to estimate the graph?

Minor points:

- In Definition 1: $\Leftrightarrow \to \Leftarrow$.

- Eq (3) can be simplified by removing the inverse.

-Below Eq. (13) the order is given by an unordered set

- I am not very convinced by the discussion of the number of valid causal orderings. Why is this important? Once the variables are found causal discovery can (in principle) be used to find the true causal structure. Also, being much smaller than all permutations is not very helpful because this number will generally be growing superexponential (at least for sparse DAGs).

---

### Official Review · Reviewer_icPp · 2023-11-01

**Soundness:** 2 fair
**Presentation:** 1 poor
**Contribution:** 2 fair
**Rating:** 5
**Confidence:** 3

**Summary:**

The work introduces a method for identifiable unsupervised representation learning, where the latent representations are encouraged to follow an ordering consistent with the (non-unique) causal ordering of a DAG. This is done by building on the work by Kivva et al., 2022, which guarantees identifiability of latent representations following a Gaussian mixture model (GMM). While (Kivva et al., 2022) prove identifiability up to affine transformations, this work shows that, by additionally enforcing a causal ordering, the ambiguity can be further reduced to a _"permutation block-diagonal transformation"_ followed by scaling and translation.

**Strengths:**

The paper attempts to provide a new method for causal representation learning, combining different results from the causal discovery and representation learning literature in a novel way.

**Weaknesses:**

1. Theorem 2 is, on its own and in its current formulation, incomplete. Supposedly, it exploits the results in (Kivva et al., 2022). However, the statement says, _"invertible mixing functions"_. For any invertible mixing functions, without further assumptions (none are stated in Thm. 2), it is possible to build counterexamples to identifiability in the i.i.d. setting based on the Darmois construction [1]. It also appears non-rigorous to state that the mixing functions are not _"identically distributed"_: the authors might be referring to the push forward through two different mixing functions of the latent distributions relative to the two models, which may generate equal distributions.

2. The mean correlation coefficient values are extremely low compared to other works on identifiable representation learning (I'm referring to MCC with respect to the ground truth variables). Even if the main contribution of the paper is meant to be theoretical, I find this result problematic. See for example the MCC results in [2, 3], the former with latent linear causal models, the latter with general mixing and general latent models but with a known causal graph: these are consistently significantly higher (~0.9) than the results reported in this paper, where it is unclear whether the reported values are better than what a non-identifiable baseline would achieve (see, e.g., [3], fig 4(b)).

3. In my understanding, the paper's main contribution is to show that, by enforcing causal ordering as described in sec. 4.1, the ambiguity up to an affine transformation in the results by Kivva et al. (2022) can be further reduced to the permutational block diagonal equivalence. This could, in itself, be considered an interesting finding. The authors' motivation lies however in CRL, so whether, and under which conditions, the latent space can be considered causal, and precisely what _"causal insights"_ may be drawn from the representations, are critically important aspects. This is, in my view, poorly discussed in the paper. For example, in [2-6], the latent models and variables are deemed causal in the sense that the modularity of causal mechanisms is exploited: i.e., _"it is possible to intervene on some of the mechanisms while leaving the others invariant"_ [3]. In turn, the assumptions underpinning the causal interpretation of the conditional distributions in this observational setting are not spelled out in this work.

4. For instance, a more in-depth discussion of latent causal graph discovery should be delved into more thoroughly. Section 4.2 reports that _"given the organised latent representations, the causal relationships can be estimated as commonly done in causal discovery"_, proceeding to cite references on the PC algorithm among others. Little discussion is dedicated to whether the assumption of causal discovery methods (such as the PC algorithm) may be expected to hold in the latent space: in particular, the faithfulness assumption---see [7]. In order to infer the DAG from (conditional independences of) the latent distribution, in a purely observational setting, it seems like the faithfulness assumption should be required; but as far as I could see, this is specified nowhere in the text. Moreover, since this assumption may be problematic in practice even for observed variables [8], a discussion of this would be required.

5. Table 1 contains some inaccuracies, and it lacks several references. For example, [9] is related to counterfactual data (it is cited elsewhere in the paper); whereas (Lippe et al., 2022) does not require counterfactual data, but rather temporal sequences: also [10] should feature in the same category. Besides (Ahuja et al., 2022), [2-6] are examples of interventional CRL (with different assumptions---e.g., linear latent SCM [2], known graph [3], linear mixing [4], genericity and latent faithfulness [5], etc).

6. Minor: The reference to (Higgins et al., 2022) on page 1 is misplaced, since said paper, which introduces a symmetry-based view on disentanglement, does not fit well with the rest of the references on _"unveiling statistically independent latent variables"_.

7. Minor: There are several typos, e.g., "invertable"->"invertable" in multiple places; "hessian"->"Hessian" on page 5

References:

[1] Hyvärinen, A., & Pajunen, P. (1999). Nonlinear independent component analysis: Existence and uniqueness results. Neural networks, 12(3), 429-439.

[2] Buchholz, Simon, et al. "Learning Linear Causal Representations from Interventions under General Nonlinear Mixing." NeurIPS 2023.

[3] Liang, Wendong, et al. "Causal Component Analysis." NeurIPS 2023

[4] Squires, Chandler, et al. "Linear Causal Disentanglement via Interventions." (2023).

[5] von Kügelgen, Julius, et al. "Nonparametric Identifiability of Causal Representations from Unknown Interventions." NeurIPS 2023.

[6] Varici, Burak, et al. "Score-based causal representation learning with interventions." arXiv preprint arXiv:2301.08230 (2023).

[7] Spirtes, Peter, Clark N. Glymour, and Richard Scheines. Causation, prediction, and search. MIT press, 2000.

[8] Uhler, Caroline, et al. "Geometry of the faithfulness assumption in causal inference." The Annals of Statistics (2013): 436-463.

[9] Von Kügelgen, Julius, et al. "Self-supervised learning with data augmentations provably isolates content from style." Advances in neural information processing systems 34 (2021): 16451-16467.

[10] Lachapelle, Sébastien, et al. "Disentanglement via mechanism sparsity regularization: A new principle for nonlinear ICA." Conference on Causal Learning and Reasoning. PMLR, 2022.

**Questions:**

- Please explain Remark 1. What does this mean in practice? What is the tradeoff which operates here? What is sacrificed by making this choice?

- In Theorem 2, what does $\hat{\tau}$ represent? How can the _"data distribution"_ have a causal ordering (that should only pertain to the latent distribution)?

- Given that the model identifiability in Thm. 2 requires two of the (non-unique) valid causal orderings, this seems to encode some knowledge on the latent DAG (as also mentioned in App. D). I would like to ask whether the authors could comment on this.

---

### Author Response · Authors · 2023-11-22
**Global Response**

# Global Response

We thank all the reviewers for their valuable feedback, we greatly appreciate the effort of reviewers to read and provide feedback for our work.

> Novelty [Reviewer 4VV5, fvDC, 4VV5]

The main novelty is the **causal ordering loss** in the latent space. The use score function as a loss function to force causal ordering is novel. Previous work on causal discovery such as Rolland et al [1] used the score for analysing the causal structure, we are encouraging it which is very different. We build the identifiability proofs on top of the great work from [2]. However, Kivva et al. [2] provide no causal interpretation, while we do. We believe that learning a causal ordering together with the representation is a novel and meaningful contribution.

> Why **causal ordering** in the latent space (permutation block diagonal) is important? [Reviewer icPp,fvDC]

1. **Causal interpretation.**: A causally ordered latent has permutational block diagonal equivalence. Our latent space has endogenous following a causal ordering. For example, a latent space $[z1, z2, z3, z4]$ would indicate that $z4$ is a leaf and $z1$ is a root node. In addition, that $z3$ could only be caused by $z1$ or $z2$, but not $z4$. These are the types of causal insights we can derive from our model.
2. **Regularisation.** Arguing from an empirical perspective, MCC is higher for coVAE (ours) than MFC-VAE [1,2] (all rest kept the same). MCC is invariant to permutation (due Hungarian matching algorithm used in its computation). The use of proposed regularization reduces the search space by 99% providing a stronger equivalence relation. This indicates that our regularisation procedure via causal ordering improves the learning of latent variables.

> Universal approximation of GMM, Assumption 5 [Reviewer fvDC, 4VV5]

The identifiability result in Kivva et al. (2022) **does** hold for arbitrary distributions. Quoting directly from Kivva et al:
*Universal approximation. Under assumptions (P1)-(F1), the model (1) has universal approximation capabilities. Any distribution can be approximated by a mixture model (2) with sufficiently many components J*

As such, a GMM with sufficiently many components can always approximate nonlinear SCMs following additive noise models (ANMs). In retrospect, Assumption 5 (GMM) does not need to be an assumption. It should be a remark.


> Clarification Theorem 2 [Reviewer icPp, 4VV5]

Theorem 2 operates under Assumptions [1, 2, 3, 4, 5] as Theorem 1. We mistakenly omitted the statement `Assumptions [1, 2, 3, 4, 5]`. We are not considering **any** invertible mixing functions. You are correct, in the general case, the mixing functions would not be identifiable.

> MCC values [Reviewer icPp]

The MCC results vary across datasets. We have nonlinear ANM in the latent space which can make the task much harder than what is considered in the previous work.

> Clarify latent causal graph discovery. [Reviewer 4VV5]

Once a causally ordered latent space is obtained, the potential causal relations between later (cause) and earlier (effect) nodes can be pruned with a feature selection algorithm (e.g. Buhlmann et al. (2014)) to yield a graph which is naturally directed and acyclic without further optimisation.
Indeed, the reference to the PC algorithm should not be there. We would need further assumptions. We will correct this.


> Number of valid causal orderings. Why is this important?  [Reviewer fvDC]

Constraining the space of solutions to possible causal ordering is the main advantage, allowing causal interpretation of the latent space. However, reducing the number of possible combinations enables stronger regularisation, improving learning. See global comment.


> Clarify Proofs of Appendix A [Reviewer fvDC, 5aVx]

The distribution mixture component of the GMM is not a random variable in the SCM. Based on equation 9, if a latent space is ordered, the mean and variance of each of its dimensions should be ordered accordingly.


> Clarify Causal Ordering loss (Proposition 1) [Reviewer 5aVx]

The equation is correct. The loss function's denominator is summed from $i$ to $d$. Let us expand, at each step of the outer sum, one variable is removed. Hence, a $i = 0$ the denominator goes from $0$ to $d$. Then $0$ is removed ($i = 1$). When $i = 1$, the denominator goes from $1$ to $d$. This happens because we encourage ONE leaf at a time. In other words, for each term of the outer sum, the $ith$ variable in the latent vector is encouraged the become a leaf w.r.t all the other variables between $i+1$ and $d$.



[1] Rolland, Paul, et al. "Score matching enables causal discovery of nonlinear additive noise models." International Conference on Machine Learning. PMLR, 2022.

[2] Kivva, Bohdan, et al. "Identifiability of deep generative models without auxiliary information." Advances in Neural Information Processing Systems 35 2022.

---

### Meta-Review · Area_Chair_JWpR · 2023-12-08

**Metareview:**

This submission was reviewed by several experts in the area of causal representation learning. During review, several issues related to the presentation (in particular, the correctness of the theorems as stated) as well as weak empirical results. As pointed out by a reviewer, in the abstract alone there is a misleading claim that "so far, provably identifiable methods rely on: auxiliary information, weak labels, and interventional or even counterfactual data",  which is false since there are numerous identifiability results that don't make these assumptions and use observational data only (along with various model assumptions). Finally, the results appear incremental compared to existing work. Unfortunately I cannot recommend acceptance in its current form.

**Justification For Why Not Higher Score:**

See meta-review

**Justification For Why Not Lower Score:**

N/A

---

### Decision · Program_Chairs · 2024-01-16

Reject